# Exploring the Use of Cold Atmospheric Plasma for Sound and Vibration Generation

**DOI:** 10.3390/s24113518

**Published:** 2024-05-30

**Authors:** Nasser Ghaderi, Navid Hasheminejad, Joris Dirckx, Steve Vanlanduit

**Affiliations:** 1InViLab Research Group, University of Antwerp, Groenenborgerlaan 171, 2020 Antwerp, Belgium; steve.vanlanduit@uantwerpen.be; 2SuPAR Research Group, University of Antwerp, Groenenborgerlaan 171, 2020 Antwerp, Belgium; navid.hasheminejad@uantwerpen.be; 3Laboratory of Biomedical Physics (BIMEF), University of Antwerp, Groenenborgerlaan 171, 2020 Antwerp, Belgium; joris.dirckx@uantwerpen.be

**Keywords:** cold atmospheric plasma, ultrasound sensor, scanning Laser Doppler Vibrometer, acoustic wave visualization, mechanical excitation

## Abstract

In this study, we investigate the potential of cold atmospheric plasma (CAP) as a non-contact excitation device, comparing its performance with an ultrasound transmitter. Utilizing a scanning Laser Doppler Vibrometer (LDV), we visualize the acoustic wavefront generated by a CAP probe and an ultrasound sensor within a designated 50 mm × 50 mm area in front of each probe. Our focus lies in assessing the applicability of a CAP probe for exciting a small polymethyl methacrylate (PMMA) sample. By adjusting the dimensions of the sample to resonate at the excitation frequency of the probe, we can achieve high vibrational velocities, enabling further mechanical analysis. In contrast with traditional vibration excitation techniques such as electrodynamical shakers and hammer impact excitation, a plasma probe can offer distinct advantages without altering the structure’s dynamics since it is contactless. Furthermore, in comparison with laser excitation, plasma excitation provides a higher power level. Additionally, while pressurized air systems are applicable for limited low frequencies, plasma probes can perform at higher frequencies. Our findings in this study suggest that CAP is comparable with acoustic excitation, indicating its potential as an effective mechanical excitation method.

## 1. Introduction

Cold atmospheric plasma (CAP) has been utilized across a diverse array of applications such as surface activation before the processing steps like gluing, painting, printing, casting, foaming, coating, or siliconizing. An increase in wettability, printability, or adhesion has been achieved on polymers [1,2] and ceramics [3,4]. In addition to their applications in industrial production processes, low temperature or cold atmospheric pressure plasmas are also instrumental in some other fields such as biology, genetics, and medicine [5,6].

This paper demonstrates that plasma can generate sound waves and vibrations within a material. It examines two key aspects: First, in the section wavefront visualization, it investigates the generation of sound waves. Second, it explores the potential of CAP as an excitation device for inducing mechanical vibrations, with the aim of facilitating further modal analysis. Throughout both sections, comparisons are made with an ultrasound sensor to evaluate the effectiveness of CAP in generating sound waves and vibrations.

### 1.1. Wavefront Visualization

Different physical characteristics of cold atmospheric plasma and its applications have been explored in [7,8,9]. However, a gap remains regarding the visualization of the emitted sound wave. This part focuses on the visualization of the acoustic wave emitted by the plasma probe, which can be used for understanding its functionality by looking at the propagated sound wave. Utilizing a scanning Laser Doppler Vibrometer (LDV), we visualize the pressure wavefront of the acoustic wave generated by the plasma probe and compare it with the sound wave of the ultrasound sensor. Through the LDV, we capture the acoustic wavefront within a square field area of 50 mm × 50 mm for both the plasma probe and the ultrasound sensor.

The utilization of the LDV for visualizing acoustic wavefronts presents several notable advantages. First and foremost, it facilitates the scanning of a region of interest with a very high spatial resolution and with a rapid point scan rate in the order of milliseconds. Furthermore, the LDV offers a non-invasive approach, ensuring minimal disruption to the acoustic field during the scanning process. The underlying principle governing LDV measurements lies in the Doppler effect, which manifests when laser light interacts with a moving target, causing a shift in the frequency of the scattered light. A comprehensive two-dimensional measurement of the acoustic field can be attained through systematic scanning across the designated area with a rigid reflector positioned behind the sound wave. The LDV has been widely employed for sound wave visualization, called refracto-vibrometry in numerous research endeavors [10,11,12]. However, its application in conjunction with a plasma probe marks a novel exploration in this study. In this method, which is called refracto-vibrometry, the LDV quantifies changes in the refractive index of a medium resulting from pressure fluctuations. The relationship between the pressure change in the sound wave field and the measured velocities with the LDV for every individual scanning point is expressed in the following Equation (Equation 1) [13]:(1)$$\Delta P=\frac{1}{2.68\times 10^{-9}}\frac{n\Delta a}{a}=\frac{1}{2.68\times 10^{-9}}\frac{n}{a}\frac{V_{\mathrm{LDV}}}{2\pi f}$$
where ΔP is the pressure change in the sound field in Pa; *n* is the optical refractive index of air related to temperature, pressure, and relative humidity [14] (at 1 atm of pressure, 20 °C temperature, and a relative humidity of 50%, *n* is 1.0002714); $V_{\mathrm{LDV}}$ is the measured velocity with the LDV in m/s; *f* is the propagating frequency of the sound wave in Hz; and *a* is the length of the laser in m that goes through the fluctuating pressure area. While this equation enables the calculation of the pressure field of the wave, our study does not center on this aspect. The fundamental principles underlying this measurement technique are thoroughly explored and elucidated in detail in the references [13,15].

### 1.2. Mechanical Excitation

Cold atmospheric plasma probes have emerged as notable instruments across various disciplines, drawing attention for their capability to generate localized plasma discharges. Despite this interest, the full scope of potential applications for these probes, particularly non-contact mechanical excitation, has remained largely unexplored. Notably, the vibrations generated by plasma can serve as an excitation source for modal analysis, similar to other methods like laser excitation [16], pressurized air systems [17], or electrodynamic shakers [18]. Each method offers distinct advantages and limitations. For instance, laser excitation provides precise control but may lack sufficient power, while pressurized air systems are effective only at limited low frequencies. In contrast with these approaches, CAP probes offer several advantages in mechanical excitation. Unlike electrodynamic shakers or hammer impact excitation, which can alter a structure’s dynamics due to physical contact, CAP probes provide non-contact excitation, preserving the integrity of the system under investigation. Moreover, CAP excitation offers higher power levels compared with laser excitation, enabling a more robust mechanical stimulation.

The mechanical excitation part investigates the unexplored potential of CAP as a mechanical excitation method, presenting a comparative analysis with a traditional ultrasound sensor. Through experimental investigation, we aim to showcase the feasibility and efficacy of utilizing CAP for non-contact mechanical excitation. By shedding light on this innovative approach, we anticipate unlocking new opportunities for CAP probes, particularly in real-time modal analysis during surface treatment processes.

This paper is structured as follows: In Section 2, a comprehensive overview of the experimental setups for wavefront visualization and mechanical excitation is provided. For **Wavefront Visualization**, a He–Ne PSV400 Scanning LDV from Polytec alongside a CAP probe and an ultrasound sensor is utilized for comparative testing. The plasma probe, specifically a CeraPlas™ F type, is operated at a frequency of around 51.75 kHz, and the ultrasound sensor employed, a CUSA-TR60-06-2200-W68 model from CUI Devices, was chosen to operate at the same frequency to compare it with the plasma. Additionally, **Mechanical Excitation** describes the selection of polymethyl methacrylate (PMMA) as the sample material, the experimental setup, and the methodology for exciting the sample at its resonant frequency using both the CAP probe and the ultrasound sensor. This section includes detailed explanations of the data acquisition process, signal processing techniques, and analysis methods utilized in the experiments. In Section 3, the outcomes of the experiments on wavefront visualization and mechanical excitation are analyzed, comparing the performance of the CAP probe with that of the ultrasound sensor. Through the comparison and interpretation of the data, the effectiveness of CAP probes in generating sound waves and mechanical excitation is elucidated. Finally, Section 4 summarizes the findings and underscores the significance of the research in advancing the understanding and potential application of CAP probes in real-time modal analysis during surface treatment processes.

## 2. Materials and Methods

We detail the experimental setup for both wavefront visualization and mechanical excitation experiments.

### 2.1. Wavefront Visualization

The experimental setup includes a He–Ne PSV400 Scanning LDV from Polytec (Karlsbad, Germany), a CAP probe, and an ultrasound sensor for testing. The CAP probe is a CeraPlas™ F type (Munich, Germany) and operates at an adjustable operational frequency of around 51.75 kHz. By “adjustable frequency”, it means that the probe has an integrated input voltage frequency control. To obtain a high voltage transformation ratio, the input voltage signal must have a frequency equal to the second resonant frequency of the PT. However, different factors such as PT temperature and electrical load [19] cause variations in the speed of sound and, consequently, in the resonant oscillation frequency. To sustain the resonant conditions, the excitation frequency must be permanently adjusted to the changing resonant conditions.

The used CAP probe utilizes a piezoelectric transformer (PT) of the Rosen type, which operates based on the dielectric barrier discharge (DBD) principle. The PT is constructed from a multilayer lead–zirconate–titanate (PZT) structure. When driven at its resonant frequency, the PT generates high voltage at its tip. This high voltage creates an electric field that exceeds the breakdown field of air, initiating micro discharges. These micro discharges form bursts of plasma at the PT tip, generating the cold atmospheric pressure plasma. (For complete details of plasma production, see [7].) The most important physical parameters of the used plasma probe are summarized in Table 1, where $L_{\mathrm{PT}}$, $w_{\mathrm{PT}}$, and $d_{\mathrm{PT}}$ represent the length, width, and thickness of the plasma probe, respectively. It is important to mention that the plasma probe also generates ozone during operation, which allows for disinfection and sterilization applications. However, in experiments like that in our study, ensuring proper ventilation of confined working spaces becomes essential to mitigate ozone levels and maintain a safe working environment.

The ultrasound sensor is a CUSA-TR60-06-2200-W68 model from CUI Devices (Lake Oswego, OR, USA). Its resonant frequency is around 48.7 kHz, but since we want to compare it with the plasma probe, we used a 10-volt sine signal at 51.750 kHz as its input. The specifications of the used ultrasound transceiver are summarized in Table 2, where *D* and *H* represent the diameter and height of its cylindrical shape, respectively.

To achieve a high signal-to-noise ratio (SNR), a reflective tape is utilized at the back of the probe tip, and to mitigate the possible vibrations induced by the acoustic field, the reflective tape is affixed to a heavy rigid block. The measurement area targeted by the laser is situated in front of the transducer in a vertical plane parallel to the reflective tape, as depicted in Figure 1. A point-to-point scan covers the total squared area in front of the device’s tip, consisting of 5151 points: 101 points in the x-direction (equivalent to 50 mm) × 51 points in the y-direction (equivalent to 50 mm).

### 2.2. Mechanical Excitation

The material selected for these experiments is polymethyl methacrylate (PMMA), commonly referred to as cast acrylic. The PMMA sample, sourced from IMATEX N.V. (Schoten, Belgium), was cut to dimensions (4.1 mm in thickness, 11.8 mm in width, and 20.55 mm in length) using a laser cutter. We chose the length based on [22], adjusting it proportionally according to different frequencies between their experiment and the exciting frequency in ours. In principle, it utilizes Equation (Equation 2), where *L* represents the sample length, *E* denotes the modulus of elasticity, and ρ signifies the material density. This ensures that the first longitudinal natural frequency (*f*) closely matches the excitation frequency of the plasma probe and ultrasound sensor used in our experiments. Given that the value of *E* at 51.75 kHz is not available, utilizing the value corresponding to 20 kHz from prior research [22], rather than relying on datasheet values associated with frequencies near zero, is appropriate.
(2)$$L=\frac{1}{2f}\sqrt{\frac{E}{\rho }}$$

To measure surface vibrations on the sample, we utilized the Polytec PSV 400 Scanning LDV. Positioned at a distance of 40 cm from the sample, the LDV emitted a laser beam angled at 25° with respect to the sample surface at its midpoint, as illustrated in Figure 2. This configuration facilitated a precise measurement and a subsequent correction of in-plane vibrations.This is only possible if the out-of-plane vibration of the sample is assumed to be negligible. For a comprehensive measurement on the surface, we employed a rectangular grid comprising 7 × 57 measurement points, covering an area of 11.8 mm by 20.55 mm on the sample surface. This grid configuration provided a full-field view of surface vibrations, enabling a thorough examination and understanding of the sample’s dynamic behavior.

The PMMA utilized in this study was in its natural transparent state. Consequently, to prepare it for LDV measurements, the sample’s surface underwent a painting process using a white powder spray paint, Ardrox^®^ 9D1B aerosol, sourced from VECOM N.V. (Ranst, Belgium). This guarantees a high level of reflection for the emitted laser onto the sample surface, leading to a high SNR in the measurements. Additionally, as the PMMA sample is designed to oscillate longitudinally at its first harmonic resonant frequency, exhibiting minimal longitudinal displacement at the middle of its length, a small clamp is 3D-printed to secure the sample at the middle, as depicted in Figure 2. This measure helps minimize the impact of boundary conditions and reduce the damping of the oscillation, ensuring a configuration similar to that of a free beam.

Figure 1 and Figure 2 depict the setup for the plasma probe. In the comparative experiment, the ultrasound sensor is exactly positioned at the location of the piezoelectric transformer’s tip of the plasma probe.

## 3. Results and Discussion

### 3.1. Wavefront Visualization

The wavefronts generated by the CAP probe using LDV measurements are depicted in Figure 3a. The colors in Figure 3a represent the real parts of the FFT signal from measured velocity vibrations at a frequency of around 51.75 kHz. This allowed us to clearly identify the trend of the acoustic wavefronts as a function of the distance. In this representation, points on the red side of the color bar indicate compression, while those on the blue side denote rarefaction of the sound wave. Figure 3b shows the magnitude of the FFT signal for the measured vibrations. Additionally, the plasma probe position is shown on the left side of the plot. Since the plasma probe has a frequency control, the input signal was varying slightly (about ±30 Hz) around the operational frequency. Therefore, the results are plotted for the superposition of seven frequency points: 51,725 Hz, 51,737.5 Hz, 51,750 Hz, 51,762.5 Hz, 51,775 Hz, 51,787.5 Hz, and 51,800 Hz. To combine the real parts shown in Figure 3a, the real parts of those seven frequency points are simply added together. Then, for the overall amplitude shown in Figure 3b, the amplitude of those frequency points is calculated using the root mean square (RMS) formula. The frequency resolution in this experiment was 12.5 Hz, determined by conducting measurements for each point over a duration of 80 ms with a sampling rate of 204,800 Hz and utilizing 6400 lines for FFT analysis.

Figure 4 shows similar plots for the acoustic wave from the ultrasound sensor at a fixed frequency of 51.75 kHz excited with a sine input with an amplitude of 10 volt. One can note the calculation by considering the relationship between the span length and the number of periods present within it. Using the relationship between wavelength, speed, and frequency ($\lambda =\frac{C}{f}$) where λ denotes the wavelength, *C* represents the speed of sound, and *f* is the propagating frequency. Therefore, λ=346m/s/51,750Hz=6.68mm, and so the expected number of periods within the span is 50mm/6.68mm≈7.48 periods.

The wavefront for the plasma appears less focused and smooth compared with that of the ultrasound sensor. This difference can be attributed to differences in probe structures. While the plasma is generated from the tip of a piezoelectric transformer (PT) with a section area of 6mm×2.8mm (see Figure 5), its emission is not as uniform and focused as the acoustic wave produced by the well-focused ultrasound sensor with a directivity of 60°. Additionally, the plasma probe’s operation can lead to the initiation of micro-discharges in one region, while others are still accumulating charge to reach the breakdown field. Consequently, multiple independent micro-discharges can occur during a single oscillation [7]. Therefore, the generated plasma is not completely uniform along the edges of the PT tip. This non-uniformity is due to small phase shifts within the plasma bursts occurring at distinct points along the edges of the PT tip. This effect is evident in the distribution of the magnitudes in measured velocities across frequencies. Figure 6 illustrates the comparative averaged magnitudes of vibrations measured at all scanning points for both the plasma probe (in blue) and the ultrasound sensor (in black). To facilitate a detailed comparison, the averaged magnitudes were normalized to their respective maximum values (corresponding to a frequency of 51.75 kHz) and converted to decibels. Notably, the noise level in both datasets is almost 45 dB lower than the level of measuring frequency (51.75 kHz), indicating minimal noise. However, the plasma graph exhibits wider distribution compared with the ultrasound sensor graph. This discrepancy can be attributed not only to slight changes in the operational frequency, typically within ±30 Hz of 51.75 kHz, but also to the minor phase deviations within the plasma bursts.

Additionally, depending on the quality of the manufacturing process, the generated plasma can be very weak in some directions as it can also be seen in the lower part of the plots in Figure 3 (the direction represented by a thin line). To check the repeatability of the measurements and to find the reason for having this weak pressure field in the shown direction, another measurement with the plasma probe flipped over was performed. The results showed that the pressure wave field is repeatable, and the reason for having a direction with weak acoustic energy can be attributed to the level of uniformity of the generated plasma on the edges of the PT tip. As mentioned, it is possible to have weaker plasma on some edges, and in this case, one of the edges of the probe is likely to have weaker plasma and, therefore, a weaker pressure field. Figure 7 shows the results for the second measurement, which indicates a weak direction in the top part of the field with the probe being flipped over.

### 3.2. Mechanical Excitation

Figure 8a illustrates the phase angles in scanning measurements across the entire surface of the sample. The purpose of showing this figure here is to confirm the claim of exciting the sample around its first longitudinal resonance mode. As it can be seen, the phase angle in the measured velocity data for all points on the right half-plane is 180 degrees with a delay in phase regarding the points on the left half-plane, indicating that the sample has been excited at its first resonance mode with a frequency of approximately 51,750 Hz. The magnitude of the measured velocity for the sample surface is shown in Figure 8b. These magnitudes are measured at an angle of $25^{\circ }$ with respect to the sample surface (see Figure 2). Due to the adjustment of the input frequency in the plasma (about ±30 Hz around the operational frequency), both Figure 8a,b are plotted using seven frequency points like in Section 3.1. To derive longitudinal velocities along the length of the sample, a correction is necessary. The results are presented without angle correction, as the primary aim of this research is to compare the plasma probe with the ultrasound sensor under mechanical excitation and demonstrate its capability to stimulate the sample.

Figure 8c,d are included for improved visualization and present the side view of the plots. As mentioned before, the length of the sample is 20.55 mm. However, since the laser spot, even with the maximum possible focus, has a radius of around 0.5 mm, the resulting scanning area in the *X* direction ranges from −9.75 mm to +9.75 mm in the plots.

Figure 9 displays the phase angle of the mode shape and the magnitudes of measured velocities of the scanning area for the ultrasound sensor. Comparing the results reveals that both methods are capable of exciting the sample at its resonant frequency. However, comparing Figure 8d and Figure 9d, it should be mentioned that the results obtained using the ultrasound sensor appear smoother. Of course, a few outlier points (out of all 7 × 57 scanning points) have been averaged, enhancing the visual perception with the color bar. Additionally, measurements for each point involved complex averaging over five instances of 80 ms, sampled at 204,800 Hz, and analyzed using 6400 lines for FFT analysis. As it is explained in Section 3.1, one possible reason for the smoother results with the ultrasound sensor is the difference in the focus of the excitation source. The plasma is generated from the PT tip, and is not as focused and uniform as the acoustic wave emitted by the ultrasound sensor.

The phase angles of the mode shapes reveal that both methods are capable of exciting the sample around its resonance frequency, facilitating its application as a mechanical excitation method.

As a comparison indicator, Table 3 presents the maximum measured velocity caused by changes in the pressure wave in the air for the refracto-vibrometry experiment. It also includes the maximum measured longitudinal velocity (at an angle of $25^{\circ }$) on the PMMA sample to assess the mechanical excitation power of the probes. It is worth noting that these values may vary with changes in their input level, which are ±8 V for the plasma probe and ±10 V for the ultrasound sensor during the experiments.

## 4. Conclusions

In this paper, we compared a cold atmospheric plasma (CAP) probe with an ultrasound sensor to visualize their emitted acoustic waves and assess their effectiveness in mechanically exciting a small sample, thereby enabling further processes like modal analysis. While wavefront visualization is valuable for understanding the spatial distribution of the acoustic energy emitted from the plasma probe, our study explores the application of cold atmospheric plasma in mechanical excitation. The comparison between the ultrasound and CAP revealed differences in the smoothness of the pressure sound waves. Additionally, the results of mechanical excitation indicated that plasma excitation was not as uniform as that achieved with the ultrasound sensor likely due to the less focused and non-uniform nature of the plasma source compared with the ultrasound sensor. Nevertheless, the experiments demonstrate the potential of using plasma as an excitation source, particularly evident when looking at the phase angles in the mechanical excitation experiment.

In summary, using a small PMMA sample as a case study, our experiments show the potential of the plasma probe in mechanical excitation by vibrating the sample at its resonant frequency, facilitating modal analysis. This innovative approach promises to unlock new opportunities for CAP probes in modal analysis, potentially in real time during surface treatment.

Future research can build upon the results gained from our mechanical excitation experiments using the plasma probe to find the mechanical properties of the sample. Utilizing the measured longitudinal velocities and performing stress and strain analysis could provide valuable information about the material properties at the excitation frequency. 

## Figures and Tables

**Figure 1 sensors-24-03518-f001:**
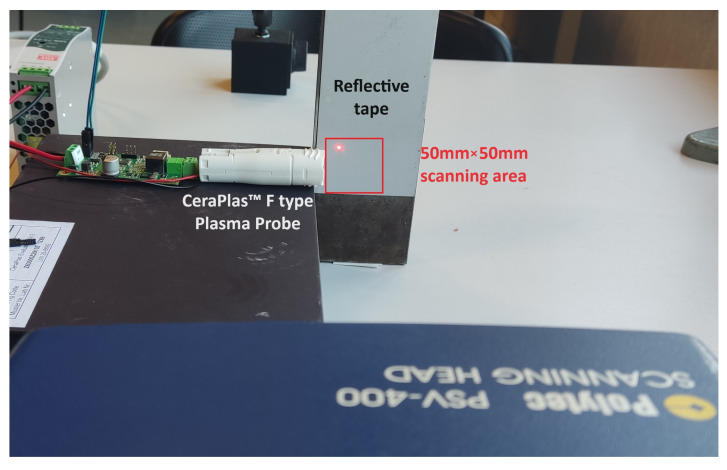
Overview of the experimental setup for wavefront visualization.

**Figure 2 sensors-24-03518-f002:**
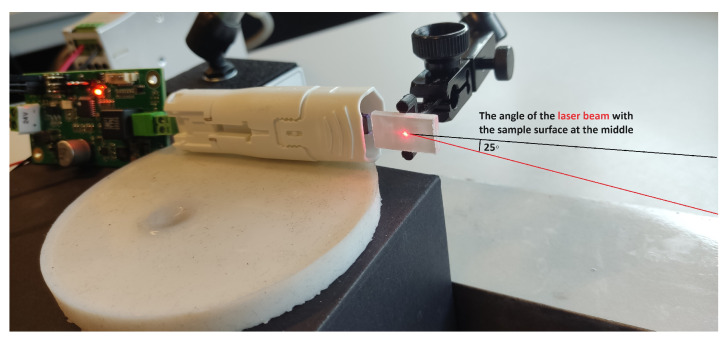
Overview of the experimental setup for the mechanical excitation of the PMMA sample.

**Figure 3 sensors-24-03518-f003:**
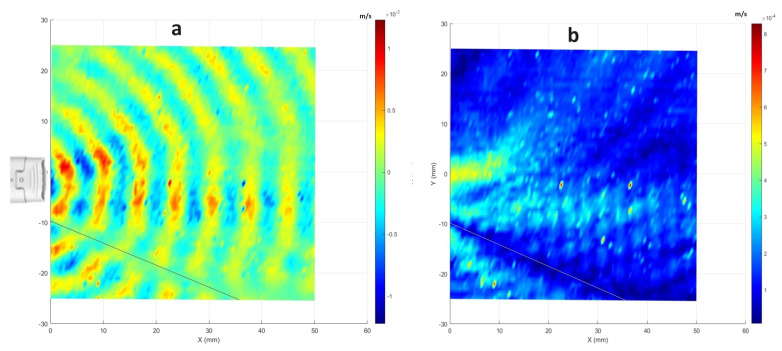
LDV visualization of the CAP probe for the excitation frequency at around 51.75 KHz: (**a**) the real part of the FFT; (**b**) the magnitude of the FFT for each scanning point.

**Figure 4 sensors-24-03518-f004:**
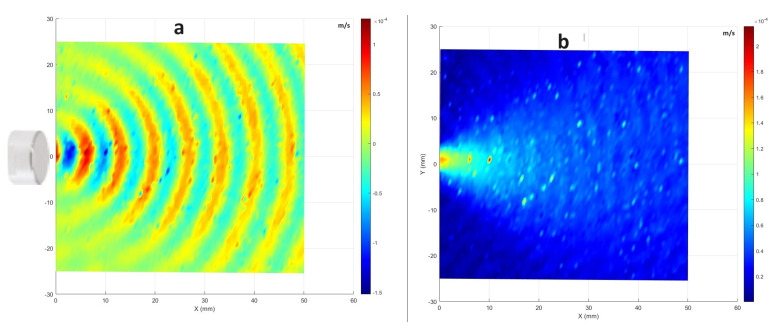
LDV visualization of the ultrasound sensor for the excitation frequency at around 51.75 KHz: (**a**) the real part of the FFT; (**b**) the magnitude of the FFT for each scanning point.

**Figure 5 sensors-24-03518-f005:**
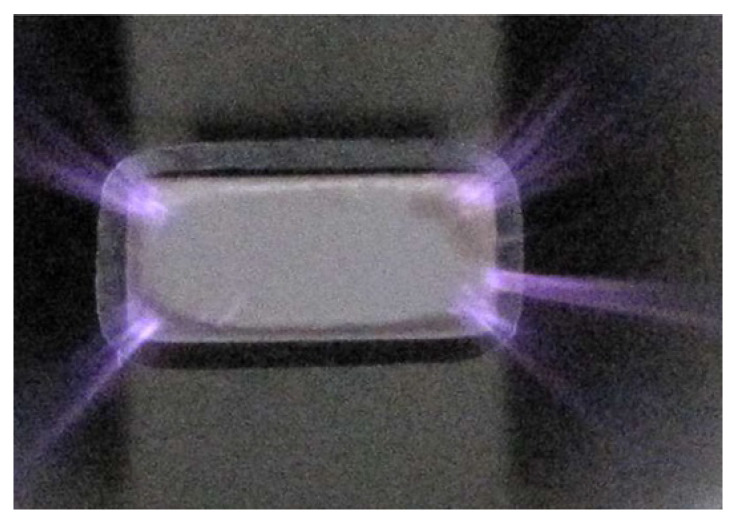
Close-up view of the piezoelectric transformer tip and the surrounding plasma generated at its edges [7].

**Figure 6 sensors-24-03518-f006:**
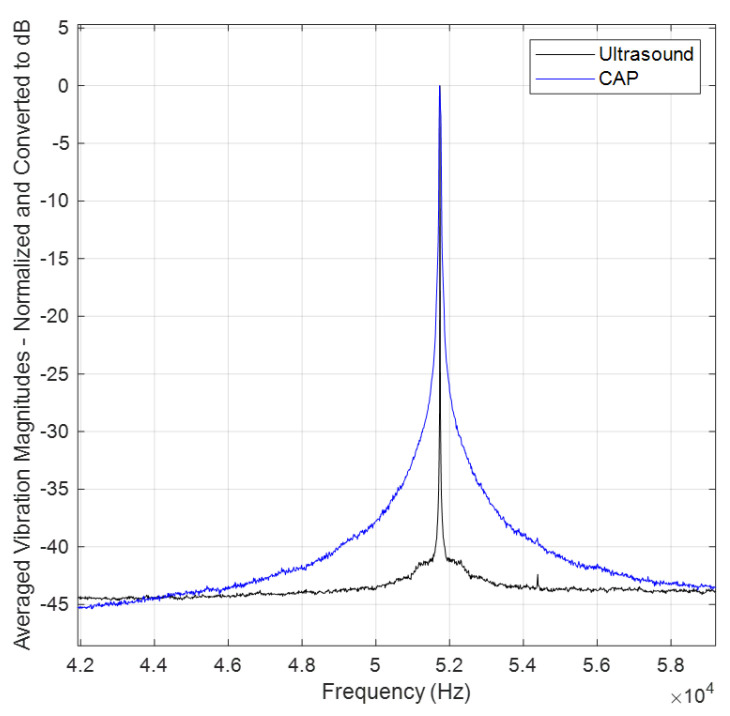
Comparison of averaged magnitudes (overall scanning points) in measured velocities across frequencies: plasma probe (blue) vs. ultrasound sensor (black).

**Figure 7 sensors-24-03518-f007:**
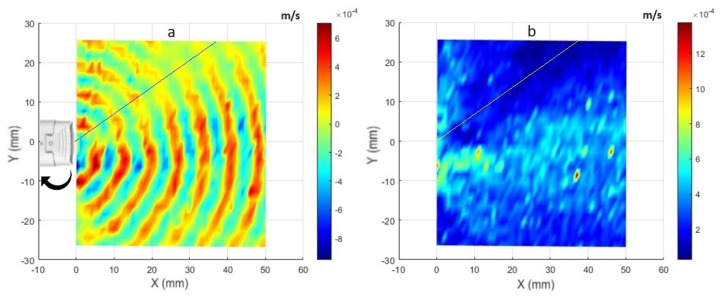
LDV visualization of the CAP probe for the excitation frequency at around 51.75 KHz with the probe being flipped over: (**a**) the real part of the FFT; (**b**) the magnitude of the FFT for each scanning point.

**Figure 8 sensors-24-03518-f008:**
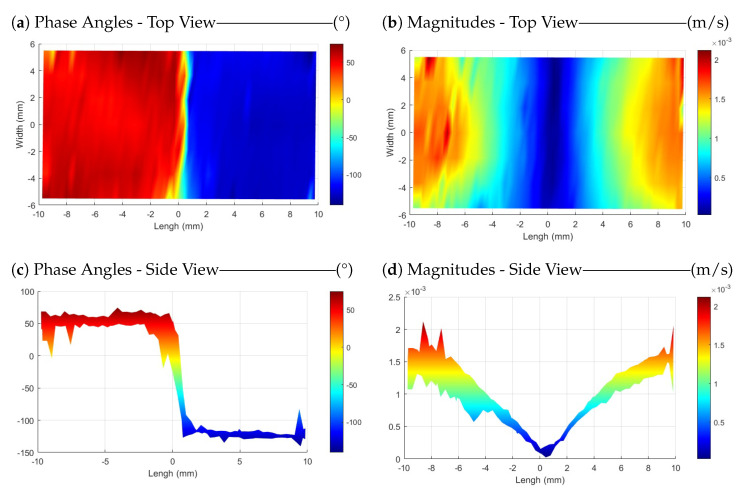
Phase angles of the mode shape (**left**) and magnitudes of longitudinal velocities (**right**) measured across the entire scanning area of a PMMA sample excited by plasma at a frequency of around 51.75 kHz.

**Figure 9 sensors-24-03518-f009:**
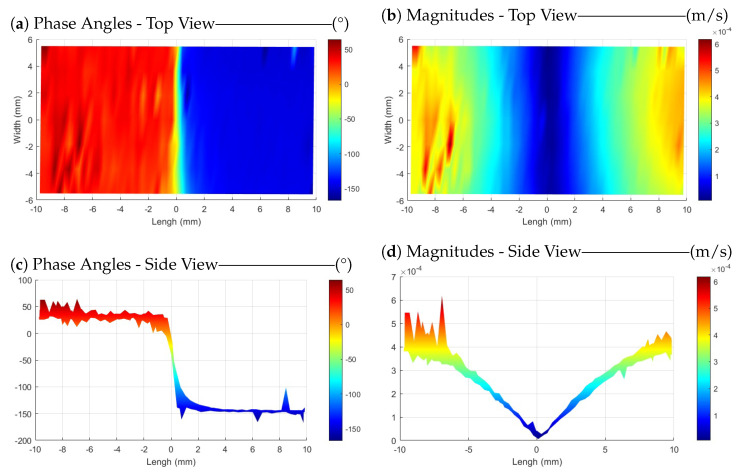
Phase angles of the mode shape (**left**) and magnitudes of longitudinal velocities (**right**) measured across the entire scanning area of a PMMA sample excited by an ultrasound sensor at a frequency of 51.75 kHz.

**Table 1 sensors-24-03518-t001:** The basic data of the cold atmospheric plasma probe [20].

Parameter Description	Parameter Value
Operating frequency (kHz)	50
Weight (g)	8.0 ± 1.6
$L_{\mathrm{PT}}\times w_{\mathrm{PT}}\times d_{\mathrm{PT}}$ (mm)	72 × 6 × 2.8
Material	PZT (lead–zirconate–titanate)
Maximum operating power (W)	8.0
Input control board voltage (V)	24

**Table 2 sensors-24-03518-t002:** The specifications of the ultrasound transceiver [21].

Parameter Description	Parameter Value
Dimensions *D* × *H* (mm)	14 × 9
Detectable Range (m)	0.3 to 6
Frequency (kHz)	48
Rated Voltage (Vp-p)	150
Directivity (°)	60
Operating Temp Range (°C)	−40 to 80

**Table 3 sensors-24-03518-t003:** Comparison of maximum amplitude in measured velocities between the ultrasound sensor and the plasma probe during wavefront visualization and mechanical excitation experiments.

Excitation Source	Max. Measured Velocity in Refracto-Vibrometry [m/s]	Max. Measured Longitudinal Velocity (at $25^{\circ }$ with Sample Surface) [m/s]
Ultrasound	$1.5\times 10^{-4}$ *	$4\times 10^{-4}$
Plasma	$10\times 10^{-4}$ *	$14\times 10^{-4}$

* With the refracto-vibrometry method, the numbers are not related to a physical velocity, but represent LDV measurements induced by changes in the pressure wave field (see Equation (Equation 1)).

## Data Availability

The data presented in this study are available on request from the corresponding author.

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
