# Peer review of "Exploring the Use of Cold Atmospheric Plasma for Sound and Vibration Generation"

_sensors, 2024, doi:10.3390/s24113518_

Round 1
Reviewer 1 Report
Comments and Suggestions for Authors
This paper introduces the potential of cold atmospheric plasma (CAP) as a non-contact excitation device and compares its performance with that of an ultrasonic transmitter.
1、 The authors did not explicitly provide the mechanism of CAP generation used in the experiments. Please include this in the revised manuscript.
2、 The results in Figure 3 show a significant amount of noise compared to the results in Figure 4. Is this noise related to the inherent instability of the plasma itself, or is it due to issues with noise? If it is due to the nature of the plasma, is there any methods can be used to resolve this? If it is due to the noise, what potential sources of noise could that be? Is there any methods to mitigate the impact of these noises?
3、 Is it possible that the noise from the CAP probe that affects the results?
4、 It is suggested that the authors conduct a quantitative analysis of the noise to enhance the application potential of this technology.
Author Response
Dear Reviewer,
Thank you for your insightful comments and suggestions. We appreciate the opportunity to improve our manuscript based on your feedback. In the attached file, we address each of your points in detail.

Reviewer 2 Report
Comments and Suggestions for Authors
Authors: Nasser Ghaderi, Navid Hasheminejad, Joris Dirckx and Steve Vanlanduit
Generation of cold plasma in an open environment is a green technology for a wide range of applications including industry, medicine and biology. Nevertheless, a use of cold atmospheric plasma for emitting acoustic waves and mechanical excitation is relatively fresh idea. In this regard, the paper by N. Ghaderi et al. is of undoubted interest.
In my opinion, this manuscript can be recommended for publication with minor corrections.
The following points should be addressed:
1. Page 2, Equation 1 [14]:
Does the equation for the pressure change in the sound field take into account relative humidity? From this point of view, expression 5 from paper [15] seems more optimal.
2. Page 6, Equation 3:
Perhaps it would be better to give the well-known relationship in the text, rather than under a separate number.
3. Page 6, lines 271-272:
“Future research can build upon the results gained from our mechanical excitation
experiments using the plasma probe to find the mechanical properties of the sample”.
It is known in advance without experiments that plasma under study is not completely uniform due to its physical nature. This can lead to error in determining the mechanical properties of the sample. Traditional acoustic excitation in this case can demonstrate a more reliable result.
Author Response

(The authors gave the same response as above.)
